

# Difference in the recruitment of intrinsic foot muscles in the elderly under static and dynamic postural conditions

Zhangqi Lai[1,*], Ruiyan Wang[2,*], Bangguo Zhou[3], Jing Chen[1] and Lin Wang[4]

[1] The Third School of Clinical Medicine (School of Rehabilitation Medicine), Zhejiang Chinese Medical University, Hangzhou, China

[2] School of Rehabilitation Science, Shanghai University of Traditional Chinese Medicine, Shanghai, China

[3] Department of Medical Ultrasound, Shanghai Tenth People's Hospital, Shanghai, China

[4] Key Laboratory of Exercise and Health Sciences (Shanghai University of Sport), Ministry of Education, Shanghai, China

[*] These authors contributed equally to this work.

Corresponding authors
Jing Chen, doctorchen1975@163.com, 20221040@zcmu.edu.cn
Lin Wang, wanglin@sus.edu.cn

## ABSTRACT

**Background**. The effect of foot, especially intrinsic muscles, on postural control and its related mechanisms remain unclear due to the complex structure. Therefore, this study aims to investigate the activation of intrinsic foot muscles in the elderly under static and dynamic postural tasks.

**Methods**. Twenty-one elderly participants were included to perform different postural tests (sensory organization test (SOT), motor control test (MCT), limit of stability test (LOS), and unilateral stance test) by a NeuroCom Balance Manager System. The participants were instructed to maintain postural stability under conditions with combined different sensory inputs (vision, vestibular, and proprioception) in SOT as well as conditions with translation disturbance in MCT, and to perform an active weight-shifting tasks in LOS. During these tasks, muscle activation were simultaneously acquired from intrinsic foot muscles (abductor halluces (AbH) and flexor digitorum brevis (FDB)) and ankle muscles (anterior tibialis, medial head of gastrocnemius, lateral head of gastrocnemius, and peroneus longus). The root-mean-square amplitude of these muscles in postural tasks was calculated and normalized with the EMG activity in unilateral stance task.

**Results**. The activation of intrinsic foot muscles significantly differed among different SOT tasks ($p < 0.001$). *Post-hoc* tests showed that compared with that under normal condition 1 without sensory interference, EMGs increased significantly under sensory disturbance (conditions 2–6). By contrast, compared with that under the single-sensory disturbed conditions (conditions 2–4; 2 for disturbed vision, 3 for disturbed vestibular sensation, 4 for disturbed proprioception), activation was significantly greater under the dual-sensory disturbed postural tasks (conditions 5 and 6; 5 for disturbed vision and proprioception, 6 for disturbed vestibular sensation and proprioception). In MCT, EMGs of foot muscles increased significantly under different translation speeds ($p < 0.001$). In LOS, moderate and significant correlations were found between muscle activations and postural stability parameters (AbH, $r = 0.355–0.636$, $p < 0.05$; FDB, $r = 0.336–0.622$, $p < 0.05$).

**Conclusion**. Intrinsic foot muscles play a complementary role to regulate postural stability when disturbances occur. In addition, the recruitment magnitude of intrinsic foot

muscles is positively correlated with the limit of stability, indicating their contribution to increasing the limits of stability in the elderly.

## INTRODUCTION

In weight-bearing activities, the human foot is the only body segment in direct contact with the ground and plays an important role in static posture and dynamic movements (*Menz, Morris & Lord, 2005*). The intricate coordination of structures within the foot is a key feature for bipedal locomotion (*Holowka & Lieberman, 2018*). In accordance with a new paradigm, the foot core system was proposed to delineate the structure and help integrate the core function of the foot (*McKeon et al., 2015*). In this system, three subsystems combine to control foot motion and stability. Bones and ligaments form foot arches as a passive subsystem that coalesce in the structural foundation of the foot and are considered as independent structures responsible for foot function (*Mc, 1955*). Foot muscles, including intrinsic and extrinsic foot muscles, play a crucial role in the functional linkage of the arches between ankle and are known as the active subsystem of the foot. Intrinsic foot muscles originate and insert on the foot and act as local stabilizers to support arches (*Soysa et al., 2012*).

Previous studies suggested that the foot contributes to human postural stability in two ways (*Menz, Morris & Lord, 2005*). One is by providing mechanical support through functional coordination between the arches and lower limb muscles. The other is by transmitting sensory information from plantar mechanoreceptors. On the basis of the paradigm of the foot core system, the function of these three subsystems are coordinated to assist in the realization of the foot's stability and activities in static posture and dynamic movement (*McKeon et al., 2015*). Indeed, the active subsystem (foot muscles) has been verified to contribute to human physical performance. Previous studies demonstrated that the toe flexor muscle strength of healthy older adults is correlated with the anterior limit of the base of support in their functional activities (*Endo, Ashton-Miller & Alexander, 2002*) and is an important contributor to postural control and mobility in the elderly (*Menz, Auhl & Spink, 2018*; *Menz, Morris & Lord, 2005*; *Mickle et al., 2009*).

Given the increasing attention to the foot, a growing number of researchers are focusing on the effect of foot muscles in human activity and their potential for facilitating functional performance. However, due to technical limitations, the function of the toe flexor muscles was often used as a proxy of the function of foot muscles, and the functional differences between intrinsic and extrinsic foot muscles were ignored. In the field of biomechanics, intrinsic foot muscles act as local stabilizers in the support of arches. A study by *Kelly et al. (2014)* showed that vertically increasing the loading of the foot caused the significant deformation of the longitudinal arch. This finding revealed that the activity of intrinsic foot muscles (abductor halluces (AbH), flexor digitorum brevis (FDB), and quadratus

plantae (QP)) increases in response to loading. Moreover, during walking and running, these intrinsic muscles act to regulate the angles of the metatarsal and calcaneus segments, thus increasing the stiffness of the longitudinal arch (*Kelly, Lichtwark & Cresswell, 2015*). Additionally, several studies have further investigated the role of intrinsic foot muscles through various methods, such as electrical stimulation activation (*Okamura et al., 2018*), fatigue induction (*Headlee et al., 2008*), and nerve blocking (*Farris et al., 2019*). Similarly, their results showed that the fatigue of intrinsic foot muscles leads to a considerable increase in foot pronation (*Headlee et al., 2008*) in the static stance, helping stiffen the distal foot and assisting pushing off during walking or running (*Farris et al., 2019*; *Okamura et al., 2018*).

Studies designed to identify the specific role of intrinsic foot muscles in postural regulation are limited. For example, an intramuscular electromyography (EMG) study (*Kelly et al., 2012*) suggested that in single-leg stance tasks, the recruitment of muscles is correlated with the medio-lateral (ML) sway of center of pressure (CoP). Recently, *Wallace, Rasman & Dalton (2018)* reported that intrinsic foot muscles show vestibular-driven muscle responses that simulate spatial transformation. Moreover, some researchers have questioned the magnitude of the role of small foot muscles in overall postural regulation (*Koyama & Yamauchi, 2017*). Although these studies provided unique insights into the function of foot muscles, most related studies focused on healthy young adults.

However, in *Kelly et al. (2012)*, a significant increase of foot intrinsic muscles was found in the single leg stance, while no significant difference was found between double leg stance and relaxed sitting. Also, the correlation between recruitment of foot muscles and CoP was only present in the single leg task (*Kelly et al., 2012*). Moreover, *Koyama & Yamauchi (2017)* found that the fatigue of foot muscles of young adults didn't induce significant change in the postural stability under double leg stance. Therefore, we suspected that for young adults with better ability of postural control, the foot muscles additionally activated to supplement postural modulation (*Ridge et al., 2022*).

Multiple physiological functions related to postural control decline with aging, such as the weakness of lower-limb muscle strength and decline of sensory integration function (*Dominguez, 2020*; *Pasma et al., 2014*). For older individuals, falls are the one of leading cause of non-accidental deaths, and widely accepted to be associated with poor postural control (*Polastri, Godoi & Gramani-Say, 2017*). Therefore, as a major public health issue, it is of great practical significance to explore the risk factors of falls and decreased postural control ability in the elderly and further explore corresponding prevention and intervention programs. Several studies has been reported that the function of foot muscles differed between the young and old adults (*Mickle et al., 2016*; *Suwa et al., 2017*; *Uritani et al., 2014*). As previously mentioned, in the elderly population, its function correlated with the postural stability and was considered as an important contributor to balance and mobility (*Endo, Ashton-Miller & Alexander, 2002*; *Menz, Auhl & Spink, 2018*; *Menz, Morris & Lord, 2005*; *Mickle et al., 2009*). However, for overall postural stability, the limited effect was found on the foot muscle for young adults, especially in the bipedal stance (*Yamauchi & Koyama, 2019*). It remains doubtful whether there are differences in the role of foot muscles in the regulation of postural control in the elderly population.

To our knowledge, a study characterizing intrinsic foot muscle activity in older adults during postural tasks has not been designed. Therefore, the purpose of our study is to investigate the muscle activity of intrinsic foot muscles under the conditions of different postural tasks, thus providing new perspectives considering foot muscle function and postural control. And it is of great practical significance to explore the potential of strengthening foot muscles to increase postural stability in the elderly. We hypothesize that the degree of foot muscles' recruitment in older adults varied across postural tasks and activation increased in response to postural perturbations.

## MATERIALS & METHODS

### Participants

A total of 21 healthy participants over 60 years of age volunteered to participate in this study. There was no gender difference in age and BMI (female: $n = 13$, age: $67 \pm 3$ years, weight: $56 \pm 7$ kg, height: $159 \pm 6$ cm, BMI: $22.3 \pm 2.9$ kg/m$^2$; male: $n = 8$, age: $70 \pm 2$ years, weight: $68 \pm 10$ kg, height: $168 \pm 11$ cm, BMI: $23.9 \pm 1.4$ kg/m$^2$). The G*POWER software 3.1 was used to calculate the sample size for the main analysis for repeated measurements ANOVA test. Based on current study design, at least 19 participants were required with setting of $\alpha = 0.05$, power $(1 - \beta) = 0.80$ and medium effect size $= 0.25$. Considering the data collection and processing, 21 participants were recruited in our study. These participants were informed of all test procedures, requirements, benefits, and potential risks before the formal test and provided written informed consent. Exclusion criteria included (1) severe cardiopulmonary disease; (2) history of lower limb injury such as trauma or ligament injuries in the past year; (3) diseases related to postural control, including but not limited to orthopedic conditions, vestibular dysfunction, Alzheimer's disease, Parkinson's disease and motor neuron disorders; and (4) dysfunction of the passive and neural subsystem of the foot (based on the Foot Posture Index-6 test and vibration perception test, respectively). The FPI-6 was performed based on palpation and observation by an experienced rater. This test has 6 items with scored on a 5-point Likert scale ranging from $-2$ to $+2$, and a total score of $+1$ to $+5$ indicates a normal-arched foot (*Redmond, Crosbie & Ouvrier, 2006*). The vibratory perception threshold was measured at the location of plantar side of the first metatarsophalangeal joint using a biothesiometer. The participants with a VPT $> 15$ V were excluded (*Young et al., 1994*). In addition, eligible participants must be able to stand and walk independently and have normal cognitive function to ensure proper understanding of the test procedure with the test of minimum mental state examination (MMSE) (*Kim et al., 2005*). This study was conducted at the Sport Medicine and Rehabilitation Center, Shanghai University of Sport, and was approved by the ethics committee of the Shanghai University of Sport (No.: 102772020RT001).

### Experimental design
### Postural perturbation task

Four main tasks with varying degrees of difficulty were used to simulate static and dynamic postural perturbation tasks by a NeuroCom Balance Manager System (version 9.3; Natus Medical Incorporated, Middleton, WI, USA). It's a reliable and widely

accepted computerized dynamic posturography test which commonly used for predicting deficit of postural control (*Broglio et al., 2009*; *Miner, Harper & Glass, 2020*). The sensory organization test (SOT), limits of stability test (LOS), motor control test (MCT) and unilateral stance test (US) were selected in this study, which used commonly as main postural perturbation tasks in previous studies (*Harro & Garascia, 2019*; *Roller et al., 2018*). Among them, SOT and US tests were conducted for static postural stability, while LOS and MCT tests were conducted for dynamic postural stability. Additionally, previous study on postural stability in our group has been published in an experimental video journal, which help to understand these tasks (*Yin et al., 2020*). In this study, the postural control test was administered by a same, experienced evaluator. This system is equipped with a support surface (two force plates) and visual surroundings that can move and offer different postural perturbations. Sampling was performed at a rate of 100 Hz. Meanwhile, muscle activity was simultaneously collected by using a wireless surface EMG system (Myon Aktos, Schwarzenberg, Switzerland). As mentioned in the Introduction section, for young adults, the foot muscles play limited role on their postural stability during static upright standing (*Yamauchi & Koyama, 2019*). EMG studies have also shown that the association between the activation of foot muscle and postural stability parameters was found only in the single-legged standing test (*Kelly et al., 2012*; *Koyama & Yamauchi, 2017*). To explore the fundamental role of foot muscles during postural regulation, we selected the SOT, MCT and LOS tests in bipedal stance as the main postural perturbation task, while the US test, a single-leg stance task, was used for the standardization (Fig. 1). In SOT and MCT, the NeuroCom Balance Manager System provided different postural disturbances, and the EMG data of the participants simultaneously collected to analyze the muscle activity under different conditions. Differently, the position of the CoP for each trial was calculated from vertical and horizontal forces captured by the force plates of the NeuroCom Balance Manager System. The procedures of these tests were described in detail to all participants before the formal measurement.

### SOT

SOT involved six different conditions of interference that combined different sensory inputs (vision, vestibular, and proprioception). The six test conditions are shown in the Table 1. In Fig. 1, this system is equipped with a support surface (in the middle of the instrument base) and visual surrounding (the colored frame). The proprioception interference was achieved by moving platform, the vestibular interference was achieved by moving surrounding, while the vision interference was achieved by closed eyes. The order of testing under the six conditions was randomized to avoid bias. The test environment was kept quiet at all times to avoid affecting the participants' postural stability. The participants were instructed to remain as stable as possible under any test condition. The participants stood on the platform facing the visual surrounding with their hands placed on either side of their body. The positions of their feet were standardized to ensure that their initial positions were the same. Three tasks were performed under each condition, resulting in 18 tasks that were randomized in order. The time for each task was 20 s. The participants

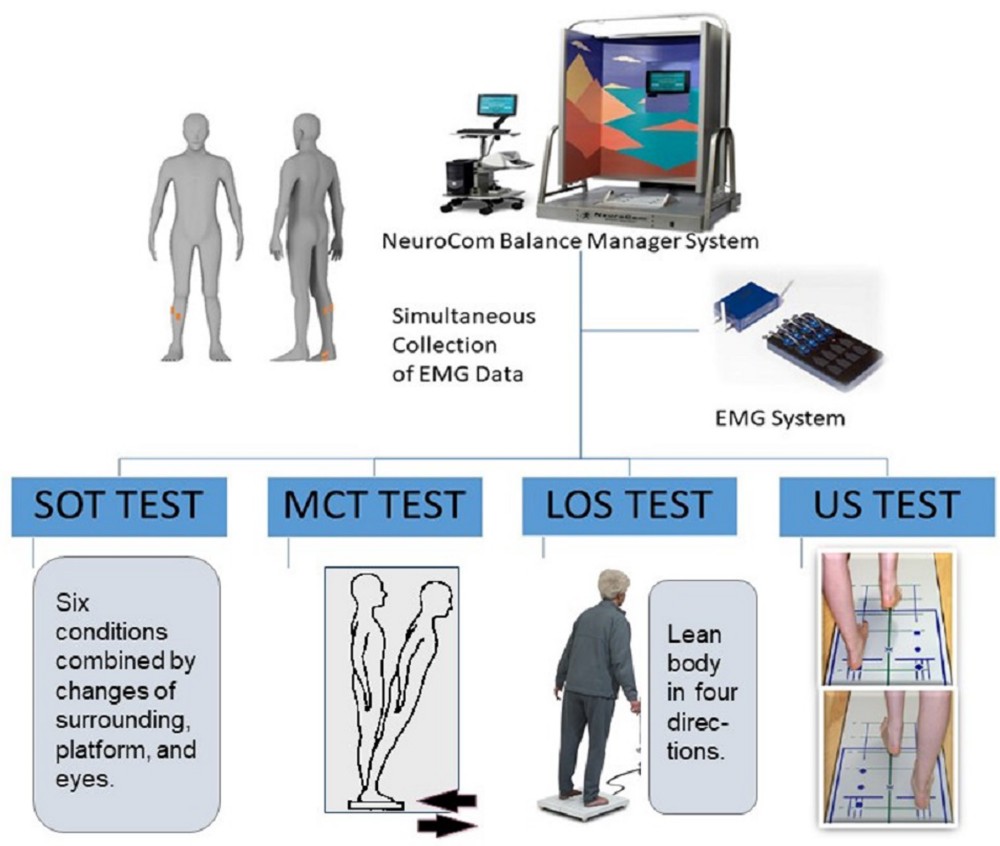

**Figure 1** Schematic diagram of experimental design.

**Table 1 Six conditions in the sensory organization test.**

| Conditions | Vision (open or closed eyes) | Vestibular sensation (moving or fixed surrounding) | Proprioception (moving or fixed platform) |
|---|---|---|---|
| One | Open | Fixed | Fixed |
| Two | Closed | Fixed | Fixed |
| Three | Open | Moving | Fixed |
| Four | Open | Fixed | Moving |
| Five | Closed | Fixed | Moving |
| Six | Open | Moving | Moving |

were provided with a rest period of approximately 15 s between each test to avoid muscle fatigue.

### LOS

LOS was conducted to quantify the participants' voluntary motor function ability. Similar to that in SOT, the participants stood on a platform and were instructed to standardize the position of their feet in LOS. Before the test, each participant was instructed to familiarize themselves with the test procedure to avoid accidents. In LOS, the participants were

required to stabilize their center of gravity in a predetermined area. They were instructed to perform an active weight-shifting task (leaning their center of gravity) as soon as possible when they heard a tone and move in each of the four directions of the body, including the forward, backward, dominant side, and nodominant side directions (*Faraldo-Garcia et al., 2016*). Each trial was conducted for 10 s. Three trials in each direction were conducted, and a rest period was provided.

### MCT

MCT was used to simulate dynamic postural perturbation task in the antero–posterior (AP) direction (forward plate, MCT-1, 2, 3; backward plate, MCT-4, 5, 6). The participants were kept in same position like SOT and LOS, and they were not informed of the direction and time of support surface's translation. The displacement amplitude was automatically adjusted according to the height of participants, and the speed followed three different speeds: low (2.8°/s), moderate (6.0°/s), and high (8.0°/s). They were required to complete three tests in the same direction and speed (totally six conditions). The data was collected during the support surface's translation. Similarly, rest time was provided to avoid muscle fatigue.

### US

In this study, US was used as a standard motion for the standardization of EMGs. In consideration of the age of the participants in our study, a single-leg stance task, *i.e.,* dominant lateral stance under the eyes-open condition, was performed three times. The participant was required to maintain a single-legged standing position with their hands at the sides of the body during the test. The nontesting leg was kept off the ground, and the participant cannot lean on or grasp any object. The EMGs of the 10 s single-leg stance task were used for subsequent processing and analysis.

## Data collection

A wireless EMG system with a 16-channel conditioning module (Myon, Aktos, Cometa Inc., Italy) was used to record several foot and ankle muscle EMGs from the dominant limb of each participant by using surface electrodes: the tibialis anterior (TA), medial head of gastrocnemius (MG), lateral head of gastrocnemius (LG), peroneus longus (PL), FDB, and AbH. We selected surface electrodes in our study rather than intramuscular electrodes (*Kelly et al., 2014*; *Kelly et al., 2012*) to avoid the risk of intramuscular wire breakage and to record foot muscle activity safely during tasks. Due to the functional connection with foot arch, the three largest of intrinsic foot muscles were commonly characterized in postural stability and activity, including AbH, FDB, and QP (*Kelly et al., 2012*; *Koyama & Yamauchi, 2017*). Among them, the QP was located deep to the FDB and its activity was usually determined by insertion of an intramuscular electrode (*Huang et al., 2019*; *Kelly et al., 2014*). Given the use of surface EMG in this study and the anatomical location of the foot muscles (mostly on the bottom of the foot), only the FDB and AbH EMGs were collected as the representation of the activity of intrinsic foot muscles. In addition, the belly of the FDB and AbH muscles was determined by using a color Doppler ultrasound system

(10 MHz, linear array, Diagnostic Ultrasound System, M7 Super; Mindary, Shenzhen, China).

After skin preparation, self-adhesive bipolar electrodes were attached over foot muscles (FDB and AbH) (*Kelly et al., 2014*; *Okai & Kohn, 2015*) and ankle muscles (TA, MG, LG and PL) in accordance with SENIAM recommendations (*Hermens et al., 2000*). The electrodes were placed along the course of muscle fibers with a distance of two cm between the centers of electrodes. By using an external synchronizer, the EMGs of the muscles were captured synchronously during all postural perturbation tasks. The EMGs were sampled at 2,000 Hz and amplified 1,000 times. The participants were asked to perform specific movements against manual resistance for EMG signal quality as follows: ankle dorsiflexion for TA, ankle plantar flexion for MG and LG, ankle valgus for PL, hallux abduction for AbH, and second-to-fifth toe flexion for FDB. Once the signals were contaminated by artefacts or were absent, the tester slightly adjusted the location of electrodes and repeated the procedure of signal quality checking. If no clear EMG signal could be recorded, the data of the participant were excluded from analysis.

## Data analysis

The EMGs of foot and ankle muscles during three postural tasks were processed by using customized software programmed in MATLAB (Version R2021a; MathWorks Inc., Natick, MA, USA). The EMGs were digitally band-pass filtered (fourth-order, zero-lag Butterworth, 10–400 Hz). In consideration of the experimental design, the EMGs in SOT, MCT, or LOT were analyzed for intratest comparisons. The time of trials was consistent under different conditions for the same task. Therefore, the EMG root mean square (RMS) was calculated for the entire duration of each postural perturbation task and the 10 s US test. The US task was utilized as a standard movement to standardize EMGs because the elderly participants were unable to complete all maximum volunteer contraction tasks. The EMGs of all trials were expressed as normalized percentage values (NRMS).

$$\text{NRMS}(\%) = \frac{\text{EMG}_{\text{rms}}}{\text{SM}_{\text{rms}}} * 100\%.$$

For LOS, the CoP path transitions in the AP and ML directions were calculated throughout the entire tasks in four directions. The following parameters were calculated to describe the movement of CoP and were analyzed by using MATLAB (Version R2021a, MathWorks Inc): total length (TL, cm), sway area (SA, cm$^2$), maximum range of AP sway (AP range, cm), and maximum range of ML sway (ML range, cm). The corresponding equations were as follows (*Koyama & Yamauchi, 2017*):

$$\text{TL} = \sum_{N=1}^{N-1} \sqrt{(\text{AP}_{n+1} - \text{AP}_n)^2 + (\text{ML}_{n+1} - \text{ML}_n)^2},$$

$$\text{SA} = \frac{1}{2} \sum_{N=1}^{N} |\text{AP}_{n+1}\text{ML}_n - \text{AP}_n\text{ML}_{n+1}|,$$

$$\text{AP Range} = \text{AP}_{\text{max}} - \text{AP}_{\text{min}},$$

$$\text{ML Range} = \text{ML}_{max} - \text{ML}_{min},$$

where $N$ was the number of CoP points, and $n$ was the CoP time series.

## Statistical analysis

Statistical analyses, including the calculation of the mean and SDs, were performed by using SPSS statistical software (version 20.0 for Windows; SPSS, Inc., Chicago, IL, USA). The Kolmogorov–Smirnov test was used to test for the normality of distribution. For the SOT and MCT tests, one-way repeated measurements ANOVA was used to determine differences in NRMS among tasks under different conditions. Mauchly's test was conducted to verify sphericity, and if the assumption of sphericity was not satisfied, the F value was adjusted in accordance with the Greenhouse–Geisser procedure. Bonferroni's *post hoc* test was applied to identify the differences among the six conditions or among different muscles. Partial eta squared was calculated to examine the effect size. For the LOS test, Pearson's or Spearman's correlation analysis was used to determine whether a correlation existed between the activation of foot muscles and the parameters of postural stability in healthy older adults. The correlation coefficient ranged from $-1$ to $1$, where $0.7 \leq |r| < 1$ represents a strong correlation, $0.3 \leq |r| < 0.7$ means a moderate correlation, and $|r| < 0.3$ indicates a weak correlation. Statistical significance was set at 0.05 for all analyses.

# RESULTS

## SOT

Table 1 shows that this test involved six different tasks, each consisting of three combinations of sensory disturbances. Statistically significant differences in AbH and FDB activity were detected among these six tasks (AbH, $F_{2.28} = 9.07$, $p < 0.001$, $\eta^2 = 0.31$; FDB, $F_{1.47} = 11.94$, $p = 0.001$, $\eta^2 = 0.37$). Further post hoc tests revealed that sensory input disturbances induced a higher recruitment level of foot muscles, as evidenced by the significantly greater NEMG of AbH and FDB under single-sensory disturbed conditions (conditions 2, 3, and 4) than under the normal condition (condition 1) and significantly greater NEMG under dual-sensory disturbed conditions (conditions 5 and 6) than under single-sensory disturbed conditions (all $p < 0.05$). In addition, no significant difference was found between single-sensory disturbed conditions or dual-sensory disturbed conditions, as illustrated in Fig. 2.

Figure 3 and Table 2 depict that the activation of intrinsic foot muscles (AbH and FDB) and ankle muscles (TA, MG, LG, and PL) increased with the difficulty of the postural control task. However, in the first three tasks (SOT 1—3), the activation of MG and LG was higher than that of AbH, FDB, TA, and PL (MG *vs* LG, $p > 0.05$ in all three tasks; MG and LG *vs* other muscle, $p < 0.05$ in all three tasks). In SOT 4, in which proprioception was disturbed, the activation of PL was lower than that of other muscles (all $p < 0.05$), whereas no significant difference was found among TA, MG, LG, AbH, and FDB (all $p > 0.05$). Moreover, the activation of PL was lowest (all $p < 0.05$) in SOT 5 (vision and proprioception disturbed) and SOT 6 (vestibular sensation and proprioception disturbed). In SOT 6, comparable levels of activation were found among AbH, TA, MG, LG, and FDB

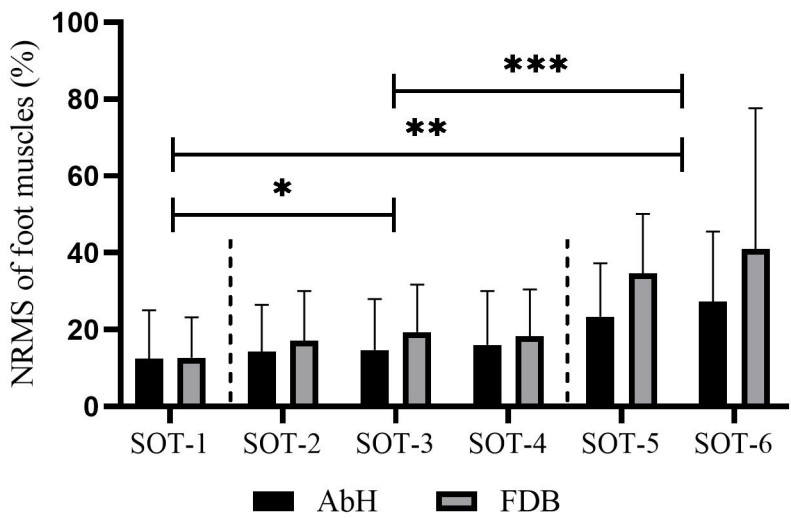

**Figure 2** **Differences in the muscle activity of intrinsic foot muscle under the six conditions of SOT.** AbH, Abductor hallucis; FDB, Flexor digitorum brevis; SOT, Sensory organization test with six different conditions; *, significant difference between single sensory disturbed conditions (conditions 2, 3 and 4) and normal condition (condition 1), $p < 0.05$; **, significant difference between dual sensory disturbed conditions (conditions 5 and 6) and normal condition (condition 1), $p < 0.05$; ***, significant difference between dual sensory disturbed conditions (conditions 5 and 6) and single sensory disturbed conditions (conditions 2, 3 and 4), $p < 0.05$.

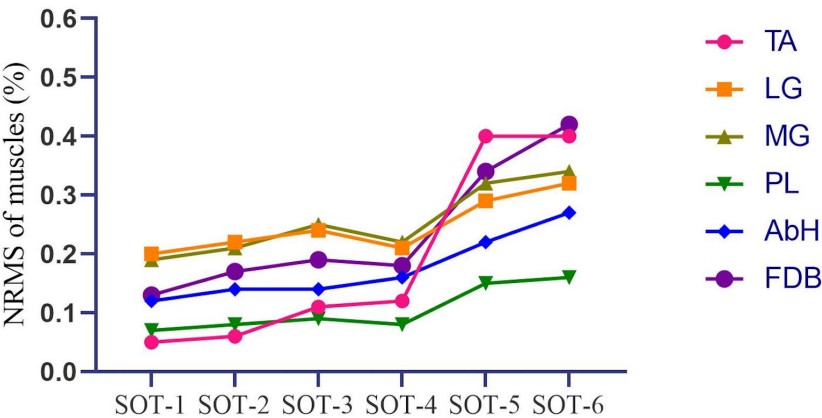

**Figure 3** **Tendency of the activation of the intrinsic foot muscles and ankle muscles in different tasks of SOT.** SOT, Sensory organization test with six different conditions; TA, Tibialis anterior; LG, Lateral head of gastrocnemius; MG, Medial head of gastrocnemius; PL, Peroneus longus; AbH, Abductor hallucis; FDB, Flexor digitorum brevis.

(all $p > 0.05$). In SOT 5, the activation of AbH was significantly different from that of PL (AbH *vs* PL, MD = 0.05, 95% CI [0.03–0.07], $p < 0.001$) and other muscles (AbH *vs* TA, MD = −0.06, 95% CI [−0.10 to −0.02], $p = 0.003$; AbH *vs* MG, MD = −0.07, 95% CI [−0.08 to −0.05], $p < 0.001$; AbH *vs* LG, MD = −0.06, 95% CI [−0.08 to −0.04], $p < 0.001$; AbH *vs* FDB, MD = −0.09, 95% CI [−0.11 to −0.06], $p < 0.001$).

**Table 2   Normalized EMG activation (% EMG-US) of intrinsic foot muscles and ankle muscles in SOT.**

|  | TA (%) | LG (%) | MG (%) | PL (%) | AbH (%) | FDB (%) | *p*-value |
|---|---|---|---|---|---|---|---|
| SOT-1 | $0.05 \pm 0.04$ | $0.20 \pm 0.12$ | $0.19 \pm 0.14$ | $0.07 \pm 0.05$ | $0.12 \pm 0.13$ | $0.13 \pm 0.10$ | <0.001 |
| SOT-2 | $0.06 \pm 0.04$ | $0.22 \pm 0.11$ | $0.21 \pm 0.13$ | $0.08 \pm 0.05$ | $0.14 \pm 0.12$ | $0.17 \pm 0.13$ | <0.001 |
| SOT-3 | $0.11 \pm 0.14$ | $0.24 \pm 0.13$ | $0.25 \pm 0.15$ | $0.09 \pm 0.06$ | $0.14 \pm 0.13$ | $0.19 \pm 0.12$ | <0.001 |
| SOT-4 | $0.12 \pm 0.18$ | $0.21 \pm 0.11$ | $0.22 \pm 0.11$ | $0.08 \pm 0.05$ | $0.16 \pm 0.14$ | $0.18 \pm 0.12$ | 0.01 |
| SOT-5 | $0.40 \pm 0.45$ | $0.29 \pm 0.09$ | $0.32 \pm 0.13$ | $0.15 \pm 0.10$ | $0.22 \pm 0.15$ | $0.34 \pm 0.16$ | 0.04 |
| SOT-6 | $0.40 \pm 0.39$ | $0.32 \pm 0.14$ | $0.34 \pm 0.17$ | $0.16 \pm 0.15$ | $0.27 \pm 0.19$ | $0.42 \pm 0.36$ | 0.01 |

Notes.

SOT, Sensory organization test with six different conditions; TA, Tibialis anterior; LG, Lateral head of gastrocnemius; MG, Medial head of gastrocnemius; PL, Peroneus longus; AbH, Abductor hallucis; FDB, Flexor digitorum brevis.

## MCT

In MCT, the EMG activity of foot and ankle muscles was analyzed in same direction (forward translation, MCT-1, 2, 3; backward translation, MCT-4, 5, 6). At forward translation tasks, there was a significant difference found in activation of AbH in different speeds (AbH, $F_2 = 64.34$, $p < 0.001$) (low *vs* moderate, MD $= -0.06$, 95% CI [$-0.09$ to $-0.03$], $p < 0.001$; moderate *vs* high, MD $= -0.10$, 95% CI [$-0.13$ to $-0.08$], $p < 0.001$; low *vs* high, MD $= 0.17$, 95% CI [0.13–0.20], $p < 0.001$). At backward translation tasks, compared to lower translation speeds, the activation amplitude of foot muscles (both AbH and FDB) increased statistically at higher speed (AbH, $F_2 = 16.49$, $p < 0.001$; FDB, $F_{1.37} = 25.01$, $p < 0.001$).

## LOS

In LOS, the EMG activity of the participant's foot muscles (AbH and FDB) in active weight transfer tasks was analyzed. Table 3 shows significant correlations between foot muscle activity and CoP parameters. Moderate correlations were found between the activation of foot muscles and the parameters of postural stability during weight transfer tasks on the forward side (AbH, $r = 0.45$–$0.64$, $p < 0.05$; FDB, $r = 0.34$–$0.62$, $p < 0.05$) and dominant side (AbH, $r = 0.36$–$0.47$, $p < 0.05$; FDB, $r = 0.34$–$0.64$, $p < 0.05$). Similarly, in the backward side transfer task, the NEMG of AbH and FDB was moderately correlated with the TL of CoP (AbH, $r = 0.33$–$0.38$, $p < 0.05$; FDB, $r = 0.42$–$0.43$, $p < 0.05$).

## DISCUSSION

The function of the foot core system has highly evolved for human postural stability *via* biomechanical coordination with the lower limbs and plantar sensory information transmission (*McKeon et al., 2015*; *Menz, Morris & Lord, 2005*). Previous cross-sectional studies demonstrated that in the elderly population, the strength of foot muscles is correlated with functional postural ability and mobility (*Menz, Auhl & Spink, 2018*; *Mickle et al., 2009*). Therefore, our study aimed to characterize the activation of intrinsic foot muscles in different postural tasks. Our results indicated that the recruitment magnitude of foot muscles significantly increased with the variation in sensory input interference. Moreover, we found that intrinsic foot muscles contributed considerably to harder postural

**Table 3  Correlation between foot muscle activation and CoP parameters in LOS (*r*-value).**

| Tasks | Muscles | Total Length (TL) | Sway Area (SA) | Maximum Range of Anteroposterior Sway (AP Range) | Maximum Range of Mediolateral Sway (ML Range) |
|---|---|---|---|---|---|
| Forward side | AbH | 0.57[*] | 0.64[*] | 0.56[*,**] | 0.45[*] |
|  | FDB | 0.60[*] | 0.62[*] | 0.52[*] | 0.37[*] |
| Backward side | AbH | 0.33[*] | 0.06 | 0.22 | −0.07 |
|  | FDB | 0.42[*] | 0.24 | 0.23 | 0.01 |
| Dominant side | AbH | 0.42[*] | 0.37[*] | 0.39[*] | 0.36[*] |
|  | FDB | 0.64[*] | 0.56[*] | 0.34[*] | 0.51[*] |
| Non-dominant side | AbH | 0.15 | 0.05 | 0.23 | −0.01 |
|  | FDB | 0.21 | 0.11 | 0.27 | 0.08 |

**Notes.**

AbH, Abductor hallucis; FDB, Flexor digitorum brevis; TL, Total length; SA, Sway area; AP Range, Maximum range of antero–posterior sway; ML Range, Maximum range of medio-lateral sway.

[*]Significant correlation between muscle activation and CoP parameters, $p < 0.05$, $0.7 \leq |r| < 1$, $0.3 \leq |r| < 0.7$, and $|r| < 0.3$ represent a strong, moderate, and weak correlation, respectively.

[**]Determined by Pearson's correlation analysis, and the use of Spearman's correlation analysis if the pound sign is not marked.

perturbation tasks than ankle muscles. Our data also showed the activation magnitude of intrinsic foot muscles.

In an upright position, the human body often undergoes successive small fluctuations as an unstable inverted pendulum structure. These postural sways are often accompanied by bursts of muscle activity and changes in the lengths of the lower limbs, especially plantarflexion muscles (*Tokuno et al., 2007*). With the increase in the interest in foot functions, studies were designed to investigate the relationship between foot muscle function and human postural stability (*Menz, Auhl & Spink, 2018*; *Misu et al., 2014*; *Suwa et al., 2016*; *Uritani et al., 2016*). Moreover, in the elderly, the weakness of foot muscles has been identified as a risk factor for impaired balance and functional ability (*Kusagawa et al., 2020*; *Maeda et al., 2021*; *Menz, Morris & Lord, 2005*; *Mickle et al., 2009*; *Misu et al., 2014*; *Pol et al., 2022*), suggesting that these muscles participate in postural control.

### Foot deformation and intrinsic foot muscle activity

The human foot is in a flexible state during normal standing, with the arch deforming in response to changes in posture and displacing in an anterior–posterior direction relative to the tibia (*Wright, Ivanenko & Gurfinkel, 2012*). As suggested by earlier intramuscular EMG studies, intrinsic foot muscles act as functional units, helping stabilize the toes during the off-ground phase of walking and providing resistance against the excessive pronation of the subtalar joint (*Gray & Basmajian, 1968*; *Mann & Inman, 1964*). Kelly et al. collected and analyzed the EMG activity of intrinsic foot muscles for postural control in 10 healthy participants during balance tasks (sitting, double-leg stance, and single-leg stance) (*Kelly et al., 2012*). They showed that the recruitment of AbH, FDB, and QP significantly increased with the increase in postural challenge, and the EMG amplitudes of these three foot muscles were correlated with the ML sway of the CoP in single-leg postural

tasks. Further research by the team showed that with the increase in the additional load on the foot, arch deformation significantly increased, and the recruitment of intrinsic foot muscles increased beyond specific load thresholds (*Kelly et al., 2014*). Interestingly, the extra activation of these muscles by electrical stimulation obviously countered the arch deformation caused by additional load, suggesting that intrinsic foot muscles, which may be further involved in postural control and dynamic activities, have an active function in LA biomechanics. The anatomy and stiffness of the foot, the only structure directly in contact with the ground, actively respond to changes in loading surface or postural demands *via* feed-forward or feed-back mechanisms. The coordination of intrinsic and extrinsic foot muscles is suspected to contribute to assisting in the immediate adjustment of arch stiffness in biomechanical modulation (*Caravaggi et al., 2009*; *Kelly, Lichtwark & Cresswell, 2015*; *Ker et al., 1987*; *Zelik et al., 2015*).

Moreover, postural orientation has been well accepted to affect the motor function of the lower and upper extremities (*Chan & Ng, 2008*; *Kantak et al., 2013*). For example, significant differences exist in the recruitment properties of intrinsic hand muscles in different shoulder positions (30° adduction and 30° abduction of the shoulder joint) (*Dominici et al., 2005*). Similarly, corticospinal excitability and postural activity in lower extremity muscles have also been reported to be associated (*Guzman-Lopez et al., 2015*; *Kesar et al., 2018*). *Mayorga et al. (2017)* utilized transcranial magnetic stimulation to measure the motor-evoked potentials (MEP) of AbH and subsequently reported the positive correlation between the MEP of AbH and foot arch anatomy, indicating the increased corticospinal excitability of AbH in individuals with high foot arches. Consistently, the Hoffmann reflex of AbH, which indicates alpha-motor neuron excitability, differs between individuals with flexible flatfoot and normal foot alignment (*Huang et al., 2019*). Individuals with flexible flatfoot exhibit high active contractions rather than the strength reflexes of AbH to maintain their stability.

### Intrinsic foot muscle activities under different postural conditions

On the basis of the above mentioned studies, distinguishing the specific role of intrinsic foot muscles during postural stabilization is impossible. Whether increased intrinsic foot muscle activity aims to assist in the increased stability of the foot core system to contribute indirectly to postural stability or directly help postural stability has not been clarified. In this study, the SOT test of NeuroCom system provided different postural perturbation conditions by interfering with the type and amount of sensory input (Table 1). In contrast to the participants of the above studies, our participants all used a bipedal standing position to conduct postural perturbation tasks. In addition, the MCT was applied to simulate the horizontal translation of the center of gravity, which help to reduce the changes in muscle activity caused by arch deformity. These approaches partly avoided the postural changes in foot loading and reduced changes in muscle activity caused by arch deformation. Our results showed that in the elderly, sensory disturbances, which increased postural demand, led to a significant increase in the recruitment of intrinsic foot muscles in postural control. This finding was suggestive of the active and complementary role of intrinsic foot muscles in postural regulation.

Intrinsic foot muscles maintain stability in the ML direction and work in response to ML postural perturbations in a pattern of high levels of EMG recruitment and intermuscular co-ordination (*Kelly et al., 2012*). The central postural control mechanism has been suggested to take charge of muscle activation in response to body sways (*Loram et al., 2011*; *Luu et al., 2012*). In accordance with this theory, the highly simultaneous recruitment of intrinsic foot muscles is suspected to contribute to postural perturbations (*Kelly et al., 2012*). For stability in the AP direction, the isolated activation of FBD can evoke a significant forward displacement of CoP (*Okai & Kohn, 2015*). *Wallace, Rasman & Dalton (2018)* delivered electrical vestibular stimulation to simulate different vestibular sensations, and similar to our results, their findings were suggestive of an inverse relationship between the function of intrinsic foot muscle activity and anterior–posterior forces under opposite electrical vestibular stimulations. Moreover, the authors compared the coherence function amplitude of foot muscle activity and systemic balance response with and without visual cues. Comparable results implied that small foot muscles have active postural roles in the modulation of standing postural stability.

Our study collected the EMGs of intrinsic foot muscles during active weight-shifting tasks (LOS test) in addition to those during static postural tasks. This test was performed to evaluate the participants' stability limit (*Mat, Ng & Tan, 2017*), which is commonly defined by the base of support (*Manista & Ahmed, 2012*). In response to postural disturbances within these safe limits, stability is recovered by adopting various postural control strategies, such as the ankle and hip strategy. The previous results of correlation between postural stability and foot muscles suggested the existence of a certain relationship between the function of intrinsic foot muscles (strength or morphology) and functional balance measured by the functional reach test, standing balance test, or fall risk assessment (*Menz, Morris & Lord, 2005*; *Mickle et al., 2009*; *Uritani et al., 2016*). Furthermore, we collected and quantified the trajectory of CoP during the LOS task and directly analyzed the association of intrinsic foot muscle activity with dynamic postural stability parameters. We therefore speculated that in the elderly, intrinsic foot muscles may contribute to postural stability by increasing the limit of postural stability.

### Study limitations

Although our study is an important step in determining the effect of intrinsic foot muscles on postural stability in the elderly, we concede that we included only healthy elderly volunteers with normal foot posture and plantar sensation function instead of classifying participants into subgroups in accordance with the function of the foot core system (active, passive, and neural subsystems) and a mixed-gender sample was conducted in our study which might limit the application of our findings to different populations. Future studies are needed to determine the special role of foot muscles on postural stability by controlling the arch deformation, as well as the interactions among the functions of these three subsystems in human postural stability and to confirm the contribution of factors associated with the foot core system in postural stability. In addition, the difference between the sampling frequency of the NeuroCom system (100 Hz) and that of the EMG system

(2,000 Hz) is too large to directly conduct a frequency domain analysis of intrinsic foot muscle activity and CoP trajectory.

## CONCLUSIONS

This study demonstrated that in the elderly, sensory input disturbances during a quiet upright stance can lead to an increment in the EMG of intrinsic foot muscles. Moreover, the recruitment magnitude of these muscles increased with the increase in sensory input disturbances, suggesting their complementary role in helping regulate postural stability. For dynamic postural stability in the elderly, the foot muscles' activation was increased in response to sudden perturbations, and their recruitment magnitude was positively correlated with the limit of postural stability, indicating that foot muscles assist in increasing the limits of stability. Our findings should be of interest to clinicians or physical therapists who are in search of new ideas to improve postural control and prevent falls in older adults.

### Funding

This study was supported by the funding of the National Natural Science Fund of China (No. 82202154), the Zhejiang Province Public Welfare Technology Project (No. LGF21H270007), and the Research Project of Zhejiang Chinese Medical University (No. 2022RCZXZK01). The funders had no role in study design, data collection and analysis, decision to publish, or preparation of the manuscript.

### Grant Disclosures

The following grant information was disclosed by the authors:
National Natural Science Fund of China: 82202154.
Zhejiang Province Public Welfare Technology Project: LGF21H270007.
Research Project of Zhejiang Chinese Medical University: 2022RCZXZK01.

### Competing Interests

The authors declare there are no competing interests.

### Author Contributions

- Zhangqi Lai conceived and designed the experiments, performed the experiments, analyzed the data, prepared figures and/or tables, authored or reviewed drafts of the article, and approved the final draft.
- Ruiyan Wang conceived and designed the experiments, performed the experiments, analyzed the data, prepared figures and/or tables, and approved the final draft.
- Bangguo Zhou performed the experiments, prepared figures and/or tables, and approved the final draft.
- Jing Chen analyzed the data, authored or reviewed drafts of the article, and approved the final draft.

- Lin Wang conceived and designed the experiments, analyzed the data, authored or reviewed drafts of the article, and approved the final draft.

## Human Ethics

The following information was supplied relating to ethical approvals (*i.e.*, approving body and any reference numbers):

It was approved by the ethics committee of the Shanghai University of Sport (No.: 102772020RT001).

## Data Availability

The raw measurements are available in the Supplemental Files.

## Supplemental Information

Supplemental information for this article can be found online at http://dx.doi.org/10.7717/peerj.15719#supplemental-information.

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
