# Peer review of "Difference in the recruitment of intrinsic foot muscles in the elderly under static and dynamic postural conditions"

_PeerJ, doi:10.7717/peerj.15719_

## Round 0.1 · original submission · Major Revisions

The manuscript has some drawbacks that must be addressed: i) the introduction lacks the necessary information and reasoning to justify the use of older adults; ii) in the same way, the EMG evaluation of some intrinsic muscles of the foot (FDB and AbH), what was your reason to only evaluate them? why these two muscles are representative of the entire muscle group?; iii) in my opinion and that of some of the reviewers, the statistical analysis should be performed separately (men and women) due to the scientific evidence in this regard.

Reviewer 1 ·

Basic reporting

Dear Editor,
It was a great opportunity to review this manuscript which investigated the relationships between balance and foot muscle activity in elderly. I think that the manuscript was well designed and written, but a few issues must be handled.

Experimental design

-

Validity of the findings

-

Additional comments

Dear Editor,
It was a great opportunity to review this manuscript which investigated the relationships between balance and foot muscle activity in elderly. I think that the manuscript was well designed and written, but a few issues must be handled.
Please address the few minor issues during revision.
Abstract
Lines 46-47: Please more detail about SOT, LOS and unilateral test. How did these tests perform? Did you used a device? Etc…
Lines 52-55: I think it is not necessary to mention the statistical analyzes used in the abstract section.
Lines 56-62: Please clarify what is conditions (1-6)?
Introduction
Lines 116: The authors mentioned that there are some on healthy young adults. Why do you expect this relationship studied in the elderly to be different? Please clarify.
Line 119: Please add the references which investigate the relationships between muscle function and balance?
Did you have a hypothesis or research question? Please add the end of the introduction section.
Materials and methods
How did you decide on the number of cases? Are 21 individuals sufficient for this study?
Lines 130: The participants, who had lower limb trauma, was excluded from the study. Were individuals with other orthopedic conditions such as osteoarthritis, rheumatoid arthritis or ligament injuries included in the study?
Lines 133-134: Please give more information about used test (Foot Posture Index-6 test and and vibration perception). How were these tests used as exclusion criteria? For example, how many scores according to FPI individuals were not included in the study? Please give more details.
Line 135: How was it determined that the participants had normal cognitive functions?
Line 141: Does this device have sufficient validity and reliability to assess postural control in the elderly? Who made the measurements? Were all measurements made by the same evaluator? Please clarify.
Results
OK
Discussion
OK
Figure
Ok
Tables
OK

·

Basic reporting

The font sizes are different and there are some clear typos. In addition, several of the figures are incomplete and need to be corrected.
Specific comments are shown below.

L84-
I think the font size is different from here. You should aligned font size.

L188
What is FED?
Is it a mistake for FDB?

【Table and Figure】
P24
Please make sure to keep the presentation in the table. Align the decimal point in the ordinal column. (AP Range) is to be a newline.

Experimental design

The rationale is not clear as to why the survey should be conducted in the elderly. Also, no rationale is given as to why the measurement was limited to two intrinsic muscles of the foot.
Furthermore, the measurement method is not illustrated, and it is difficult to get an image of the measurement method, so it needs to be corrected.
Specific comments are shown below.

L110
I think you need to describe about why you need to evaluate postural control tasks in the elderly.

L119-123
Why were the participants in this study elderly?
The reasons for selecting the elderly should be stated.

L146-148
Why did you set these three tasks? If you have references, please provide a rationale based on the literature.

L187-189
Your reasoning for why you chose the FDB and AbH among the intrinsic foot muscles was based on your "Given the anatomical location of the foot muscles (mostly on the bottom of the foot), only the FED and AbH EMGs were collected as a representation of the activity of intrinsic foot muscles." seems abstract.
It would be necessary to explain the rationale for FDB and AbH being representative foot muscles.

Experimental design
It is easier to understand if a figure during the assignment is shown.

Validity of the findings

I am concerned about two points: the lack of evidence for selecting the elderly and the data from this study alone is not convincing enough for consideration. Please correct them.
Specific comments are shown below.

L346
I understood that the participants conduct postural perturbation task which is to avoid postural changes in foot loading by having the participant apply a backward perturbation task, and to reduce changes in muscle activity caused by arch deformity. But you measured only muscle activity in this study. Do you have any ancillary data that would indicate that the arch has changed? I think it would be more convincing.

·

Basic reporting

The manuscript is methodologically consistent. It manages to clearly justify the research problem that is intended to be studied. The structure follows the format of the journal.

In Table 2 it is suggested to indicate the unit of measurement of the muscle activation exhibited, while in Table 3 it is suggested to specify that the values presented are the correlation coefficients (r-value). Also, add unit of measurement for the LOS test variables (Total length, sway area, AP range, ML range).

The lines above the graphs showing the differences between the SOT conditions are unclear. They fail to express what is stated in the manuscript. I suggest checking.

Experimental design

In general, the study procedures are clear. However, I do have a few comments:

- The baseline characteristics of the participants could be presented by gender (in the next item I will refer to this again).

- How was it determined that the participants had "normal cognitive function"? Was any questionnaire or scale applied to determine this?

- The experimental design item could be written in a simpler and more understandable way for the reader. The Neurocom Balance system was used to perform SOT and to measure LOS. if so, perhaps it should only indicate that SOT was performed using this equipment and that LOS was obtained from the force platform of this instrument (and provide characteristics of the instrument and the variables it measures). In data analysis, it is possible to understand that the Neurocom Balance system is the instrument that measures the CoP. However, this should have been considered in the experimental design item. In data analysis, it should only be specified how the CoP signals were processed in MATLAB (as presented at the end of this section).

- It is noted that (line 192-194) "After skin preparation, self-adhesive bipolar electrodes were attached over foot muscles (FDB and AbH) and ankle muscles (TA, MG, LG and PL) in accordance with SENIAM recommendations(Hermens et al., 2000)". Checking on http://www.seniam.org I couldn't find FDP and AbH muscles. Is there an error in the sentence?

Validity of the findings

The results are understood in the manuscript and are consistent with the conclusions. I have a question regarding the results.

- Why were men and women not analyzed separately? There are old and recent studies that point to differences in postural control between older men and women, even under evaluation conditions similar to yours (eyes closed and somatosensory alterations).

https://www.bmj.com/content/1/6056/261.abstract

https://pubmed.ncbi.nlm.nih.gov/8014390/

https://pubmed.ncbi.nlm.nih.gov/31741484/

---

## Round 0.2 · Minor Revisions

The manuscript in its current version has been considerably improved. However, the reviewers detail a couple of issues that should be addressed by the authors. Especially, those related to the reasons why foot morphology in older people should be studied and the description of the characteristics of the investigated sample.

Reviewer 1 ·

Basic reporting

Dear Editor,
Thank the authors for their careful response to my comments. In my opinion, the current version of manuscript is suitable for publication in PeerJ.
Best regards.

Experimental design

-

Validity of the findings

-

Additional comments

-

·

Basic reporting

L126-141
Information such as the relationship between foot muscles and postural stability and a description of the differences in foot muscle function between the elderly and younger adults was added, but there does not seem to be a description of why you selected elderly. For example, the elderly is prone to falls due to poor postural stability, prevention is important because falls reduce QOL and other aspects of life. In this way, I think it would convey the significance of focusing on the elderly.

Experimental design

L175-200
Even with your further explanation, I feel that the task and the reasons for the choice are not clear. Wouldn't it still be necessary to add more detailed information?

L187
I think there is little basis for why we do SOT, LOS, and MCT. Are these three tasks common as main postural perturbation tasks? If so, they could be described as such.

L248-254
The references need to be inserted in the text regarding the reasons for the selection of AbH, FDB, and QP.

Figure1
Please include scenery that will help us visualize the task motion.
For example, the SOT is done in 6 conditions, but it is difficult to understand them with only text.
It would be easier to understand by figuring this.

Validity of the findings

L175-200
I think it would be good to have a chart of SOT, LOS, and MCT measurements respectively.

Additional comments

Thanks for the revise. I think you have fixed the general content. However, It is not enough explanation that the implication of this study, the focus on the foot morphology of the elderly people. Also, the explanation of the task movement was insufficient, so I think it needs to be improved to make it easier to understand. I hope that the following points will be taken into consideration and that this research will be further developed.

·

Basic reporting

The manuscript has improved significantly. The material and method section has a new order that favors the reading and understanding of the research carried out. I only have one comment.

- In the correlation table it would be useful to add with a symbol which r values ​​were obtained with the Pearson or Spearman tests because, as can be seen in the statistical analysis, they used one of these depending on normality. Also, in this table, it would be appropriate to add that the asterisk symbol (*) represents significant correlation. Adjust to the magazine format in case of accessing the suggestions provided.

Experimental design

In general, the study procedures are clear, but there are some aspects in which it could be improved.

- Although it is mentioned that the participants were men and women, and their basal characteristics are indicated. I suggest indicating the number (n) of men and women evaluated in the research.

Validity of the findings

The results are understood in the manuscript and are consistent with the conclusions.

---

## Round 0.3 · accepted · Accept

The authors have satisfactorily addressed all central reviewers' comments and observations. This new version of the manuscript is ready to be published.

·

Basic reporting

Thanks for making the revisions to my comment. In the method section, you provided a detailed description of the task movement, but I thought it would be easier to understand if you included an image of the actual measurement situation of the task movement. However, the manuscript has improved significantly, so I will assume this manuscript will be minor revision.

Experimental design

L233-245, L246-254
The addition of Fig. 3 has made it much easier to understand, but it is difficult to get an image of the environment measuring of SOT and MCT. It would be easier to understand if you could use actual measurement situations for each of them and explain them in figures.

Validity of the findings

I have no comment.

Additional comments

I have no comment.

·

Basic reporting

No comment.

Experimental design

No comment.

Validity of the findings

No comment.

Additional comments

The manuscript has improved in all its sections. For my part, I consider that it is fit for publication.